# Search and rescue at sea aided by hidden flow structures

Mattia Serra [1✉], Pratik Sathe [2], Irina Rypina [3], Anthony Kirincich [3], Shane D. Ross [4], Pierre Lermusiaux[5], Arthur Allen[6], Thomas Peacock [5] & George Haller [7✉]

Every year, hundreds of people die at sea because of vessel and airplane accidents. A key challenge in reducing the number of these fatalities is to make Search and Rescue (SAR) algorithms more efficient. Here, we address this challenge by uncovering hidden TRansient Attracting Profiles (TRAPs) in ocean-surface velocity data. Computable from a single velocity-field snapshot, TRAPs act as short-term attractors for all floating objects. In three different ocean field experiments, we show that TRAPs computed from measured as well as modeled velocities attract deployed drifters and manikins emulating people fallen in the water. TRAPs, which remain hidden to prior flow diagnostics, thus provide critical information for hazard responses, such as SAR and oil spill containment, and hence have the potential to save lives and limit environmental disasters.

[1] School of Engineering and Applied Sciences, Harvard University, Cambridge, MA 02138, USA. [2] Department of Physics and Astronomy, University of California, Los Angeles, CA 90095, USA. [3] Physical Oceanography Department, Woods Hole Oceanographic Institution, Woods Hole, MA 02543, USA. [4] Aerospace and Ocean Engineering, Virginia Tech, Blacksburg, VA 24061, USA. [5] Mechanical Engineering Dep., Massachusetts Institute of Technology, Cambridge, MA 02139, USA. [6] U.S. Coast Guard Office of Search and Rescue, Washington, DC 20593, USA. [7] Institute for Mechanical Systems, ETH Zürich, 8092 Zurich, Switzerland. ✉email: serram@seas.harvard.edu; georgehaller@ethz.ch

In 2016, the United Nation Migration Agency recorded over 5000 deaths among people trying to reach Europe by crossing the Mediterranean Sea[1,2]. This calls for an enhancement of the efficiency of SAR at sea[3], which requires improved modeling of drifting objects, as well as optimized search assets allocation (reviewed previously[4,5]). Flow models used in SAR operations combine sea dynamics, weather prediction, and in situ observations, such as self-locating datum marker buoys[6] deployed from air, which enhance model precision near the last seen location. Even with the advent of high-resolution ocean models and improved weather prediction, however, SAR planning is still based on conventional practices that do not use more recent advances in understanding transport in unsteady flows.

Current SAR procedures[7] approach uncertainties through Bayesian techniques, turning the modeling exercise into an ensemble integration over all unknown parameters and incorporating unsuccessful searches into locating the next target. This strategy produces probability-distribution maps for the lost object's location, which, based on a list of assigned search assets, returns search plans, such as planes flying in a regular grid pattern[7]. The vast uncertain parameter space together with the continuous motion of floating objects driven by unsteady flows, however, leads to error accumulation, "making SAR planning as much art as science, where rescuers still often rely as much on their hunches as on the output of sophisticated prediction tools"[4]. Furthermore, the convergence of updated probability computations based on a selected prior and unsuccessful searches is usually a slow process, while timing is everything when lives are on the line.

In a SAR scenario, one would ideally have a simply interpretable tool based on key features of the ocean surface dynamics. Such a tool should narrow down the search area by promptly providing the most attracting regions in the flow toward which objects fallen in the water at uncertain locations likely converge. This raises the question: How can one rigorously assess short-term variabilities of material transport in fast-changing flows characterized by high uncertainties? Here, we address this question using the recently developed concept of Objective Eulerian coherent structures (OECSs)[8] from dynamical systems theory. In our context, attracting OECSs uncover hidden TRansient Attracting Profiles (TRAPs), revealing the currently strongest regions of accumulation for objects floating on the sea surface. TRAPs are quickly computable as smooth curves from a single snapshot of available modeled or remotely sensed velocity fields, providing highly specific information for optimal search-asset allocation (Fig. 1). The inset in Fig. 1 shows a migrant boat that capsized on 12 April 2015 in the Mediterranean Sea, along with a schematic TRAP attracting people in the water (PIW).

We confirm the predictive power of TRAPs in three field experiments emulating SAR situations south of Martha's Vineyard in Massachusetts USA. In the first experiment, we compute TRAPs from a submesoscale ocean surface velocity field reconstructed from remotely sensed high-frequency radar (HFR) data, and show their decisive influence on surface drifters emulating people that have fallen in water at uncertain locations. In actual SAR operations, however, HFR velocity data is generally not available in real time. We address this challenge in our second and third experiments by computing TRAPs from an ocean model velocity field that assimilates in situ experimental information. We then verify the TRAPs' role in attracting and aligning drifters and manikins, simulating PIW, released in their vicinity through targeted deployments. Our analysis reveals a remarkable robustness under uncertainty for TRAPs: even without accounting for wind-drag or inertial effects due to water–object density difference—typically uncertain in SAR scenarios—the TRAPs invariably attract floating objects in water over two-to-three hours. Such short-time predictions are critically important in SAR.

## Results

**Lagrangian transport in fluids**. Short-term variability in flow features (or coherent structures) is substantial in unsteady flows. These structures, such as fronts, jets, and vortices, continue to receive significant attention in fluid mechanics due to their decisive role in organizing overall transport of material in fluids. Such transport is a fundamentally Lagrangian phenomenon, i.e., best studied by keeping track of the longer-term redistribution of individual tracers released in the flow. In that setting, Lagrangian coherent structures (LCSs) have been efficient predictors of tracer behavior in approximately two-dimensional geophysical flows, such as surface currents in the ocean[9].

Larger-scale models and measurements of environmental flows, however, generally produce Eulerian data, i.e., instantaneous information about the time-varying velocity field governing the motion of tracers. These velocity fields can be integrated to obtain tracer trajectories, but the result of this integration will generally be sensitive to a number of factors. One such set of factors is the exact release time, release location, and length of the observation period[9]. Another major sensitivity factor is errors and uncertainties in the velocity field, which either arise from unavoidable simplifications and approximations in modeling, or from inaccuracies in remote sensing. A third source of sensitivity is the necessarily approximate nature of trajectories generated by numerical integration, due to finite spatial and temporal resolution of the velocity data, as well as to approximations in the numerical integration process. All these factors are significant in predictions for SAR purposes: in fast-changing coastal waters, uncertainties both in the available velocities, and in the release location and time are high. This has prompted the use of multiple models, stochastic simulations, and probabilistic predictions, all of which require substantial time to be done accurately, even though time runs out quickly in these situations.

**Hidden short-term attractors of floating objects**. An alternative to Lagrangian approaches is to find the instantaneous limits of LCSs purely from Eulerian observations, thereby avoiding all the pitfalls of trajectory integration. These limiting (i.e. infinitesimally small-advection-time) LCSs, predict pathways and barriers to short-term material transport until the next batch of updated velocity information becomes available. While simple at first sight, this approach comes with its own challenges, given that most classic instantaneous Eulerian diagnostics (streamlines, velocity magnitude, velocity gradient, energy, vorticity, helicity, etc.) are not objective[10], i.e., depend on the observer. As such, they cannot possibly be foolproof indicators of material transport, which is a fundamentally frame-independent concept. Indeed, different observers relying on data collected from the coast, from an airplane, from a ship or from a satellite should not come to different conclusions regarding the likely location of materials or people in the water. Yet classic Eulerian quantities would in fact give such different answers (see e.g. Fig. 3a in ref. [9] and Fig. 1 in ref. [8]). In a SAR situation, this ambiguity is a serious limitation that represents high risk.

These considerations led to the development of OECSs[8], which are objective (observer-independent) short-term limits of LCSs. Most relevant to our current setting are hyperbolic OECSs in two-dimensional flows, which are the strongest short-term attractors and repellers of material fluid elements. As such, OECSs are extensions of the notions of unstable (and stable) manifolds of a saddle point in a steady flow, which attract (and repel, respectively) fluid elements and hence ultimately serve as the backbones of deforming tracer patterns. In unsteady flows and over short times, however, saddle stagnation points lose their connection with material transport[8]. Instead, *objective saddle points*—the cores of hyperbolic OECSs—emerge, with associated attracting and repelling

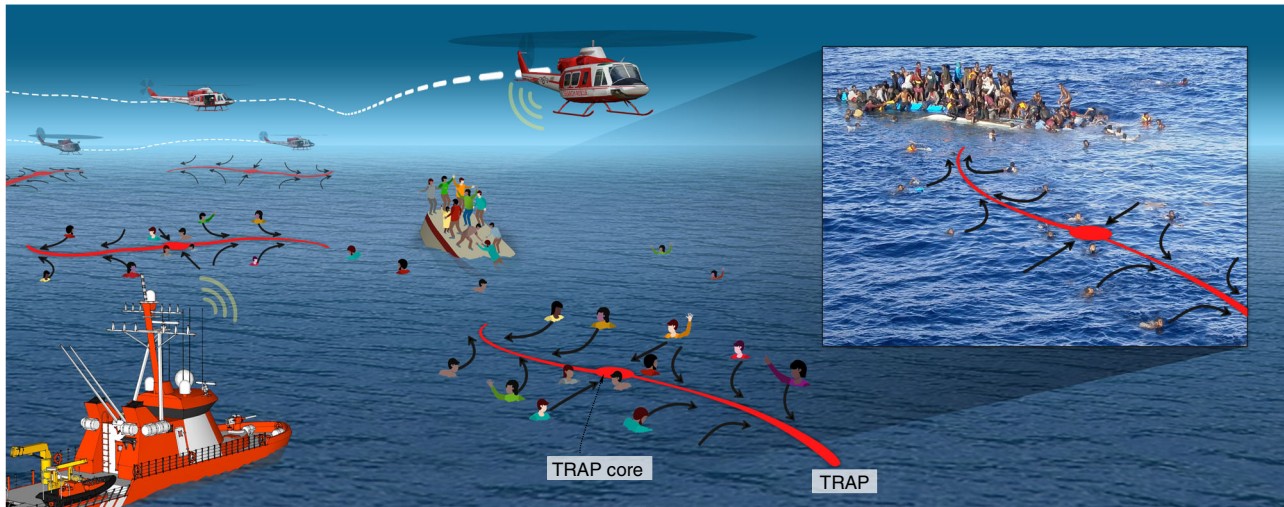

**Fig. 1 Sketch of a TRAP-based SAR operation. TRAPs (red curves) emanate from an attracting core (red dot) where their normal attraction (black arrows) is maximal.** Different TRAPs provide continuously updated and highly specific search paths. The inset shows a migrant boat that capsized on 12 April 2015 in the Mediterranean Sea along with a schematic TRAP and persons in water. Photo credit: Opielok Offshore Carrier.

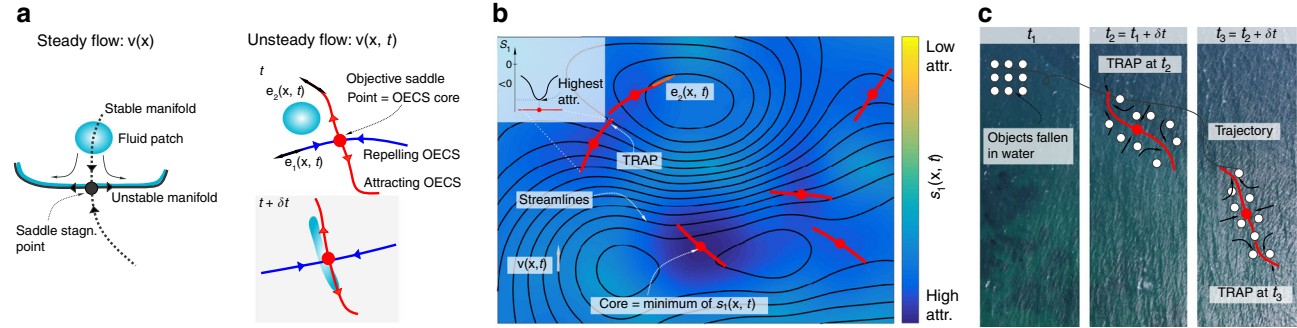

**Fig. 2 Transient attracting profiles (TRAPs). a** Deformation of a fluid patch close to a saddle stagnation point in a steady flow (left), and to an objective saddle point in an unsteady flow (right). Over short times, a fluid patch aligns with the repelling OECS, and squeezes along the attracting OECS, which both evolve over time. Attracting (Repelling) OECSs are everywhere tangent to the eigenvector field $\mathbf{e}_2$ ($\mathbf{e}_1$) of the rate-of-strain tensor, with their cores located at minima of the smallest eigenvalue $s_1$. **b** Attracting OECSs, i.e. TRAPs, in an unsteady ocean velocity data set derived from satellite altimetry data along with their normal attraction rate $s_1$ encoded in the colorbar. TRAPs are hidden to instantaneous streamlines (black). **c** Illustration of a TRAP evolving in time and attracting within a few hours floating objects whose uncertain initial locations are represented by a square set of white dots.

OECSs (Fig. 2a). In our present context, we will refer to attracting OECSs and objective saddle points as *TRAPs* and *TRAP cores*.

Unlike stagnation points in steady flows, OECSs cannot be located by inspection of a (frame-dependent) streamline configuration, but from the objective rate of strain tensor (see Methods section). As an illustration, Fig. 2b shows TRAPs in an unsteady ocean velocity data set derived from AVISO satellite altimetry (see ref. [8] for a detailed OECSs analysis of this flow). Thus, the $s_1$ scalar field along with the TRAPs provides a skeleton of currently active attracting regions in the flow along with their relative strengths. This in turn gives specific and actionable input for SAR asset allocation, such as high-priority flight paths for discovering people in the water (Fig. 1). Remarkably, such pathways remain generally hidden in streamline plots, can even be perpendicular to streamlines (Fig. 2b and Supplementary Movie 1), and exist also in divergence-free flows, as shown in Fig. 11 and the corresponding movie in ref. [8]. See ref. [8] for a detailed explanation. Figure 2c shows that TRAPs evolve over time and attract floating objects whose uncertain initial positions are represented by an array of white dots.

As Eulerian objects, TRAPs are simply computable from a single snapshot of the velocity field $\mathbf{v}(\mathbf{x}, t)$. Moreover, velocity fields used in traditional SAR planning are generally obtained from models that assimilate environmental data in the proximity to the last known position of a missing person[4,7]. This represents a further challenge to Lagrangian prediction methods, as much of their trajectory forecasts tend to leave the domain of reliable velocities and hence have questionable accuracy. In the Supplementary Fig. 4, we illustrate this effect, showing that Lagrangian methods provide only partial coverage when velocities are available over a finite-size domain. A TRAP-based analysis is, therefore, not only faster but provides complete coverage by exploiting the most accurate velocity data. In the Supplementary Note 1, we provide a rough estimate of the computational time gain of a TRAP-based SAR planning compared to the one currently in use.

Finally, owing to the structural stability of their construction[8], TRAPs necessarily persist over short times and are robust to perturbations of the underlying velocity field. In the Supplementary Methods, we show that the sensitivity of TRAPs to uncertainties is typically lower compared to those of trajectory-based methods (Supplementary Fig. 1). This makes TRAPs a trustworthy now-casting tool for material transport, one that is resilient under uncertainties in initial conditions and other

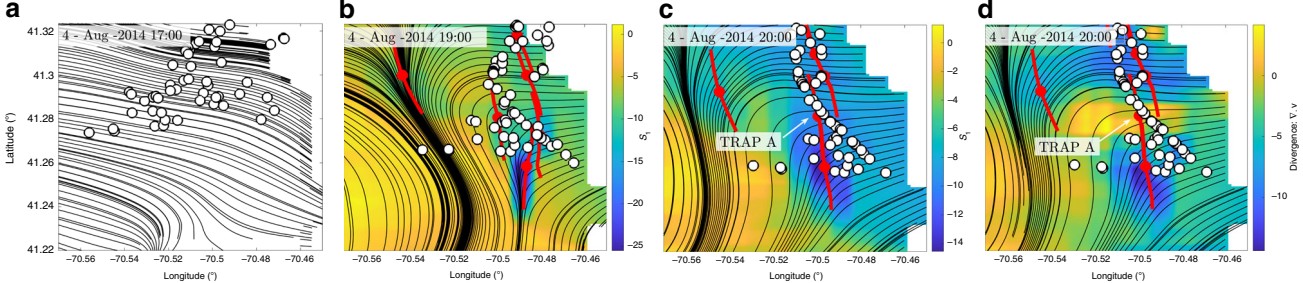

**Fig. 3 Field experiments tools and area of interest. a** The domain of the ocean field experiments is located south of Martha's Vineyard, where the ocean surface sub-mesoscale velocity, remotely sensed from high-frequency-radar (HFR) measurements as described in ref. [11], is available within the hatched black polygon. The black rectangle represents the area of interest of the 2014 field experiment. **b** Tioga WHOI vessel, CODE surface drifter, whose GPS-tracked position will be marked with a white dot, and OSCAR Water Rescue Training manikins whose GPS-tracked position will be marked with a magenta triangle. **c** Photo illustrating a drifter and a manikin in water during the 2018 experiment. A drone-based video of the 2018 field experiment is available here[17].

**Fig. 4 2014 experiment. a** Region bounded by the black rectangle in Fig. 3a showing drifter positions (white) at the beginning of our analysis (4 August 2014 at 17:00 EDT), along with the instantaneous streamlines (black) from HFR velocity. **b** and **c** TRAPs (red curves), whose normal attraction rate $s_1$ (the more negative the more attracting) is encoded in the colorbar, along with instantaneous streamlines and drifters' position, at 19:00 and 20:00 EDT. The time evolution of drifter positions along with TRAPs and velocity streamlines is available as Supplementary Movie 1. **d** Same as **c** with the colorbar encoding the horizontal divergence field ($\nabla \cdot \mathbf{v}$). TRAP A in panels **c** and **d** strongly attracts the drifters despite being in a region of positive horizontal divergence. The colorbars unit is day$^{-1}$.

unknown factors, such as the inertia of a drifting object or windage effects.

**Ocean field experiments.** Here, we show how TRAPs accurately predict short-term attracting regions to which objects fallen in water at uncertain nearby locations converge in ocean field experiments carried out south of Martha's Vineyard. Figure 3 shows the location of the experiments and the tools we used. In our first experiment, we compute TRAPs from ocean-surface sub-mesoscale velocity derived from high-frequency-radar (HFR) measurements available over a uniform 800 m × 800 m grid spanning [−70.7979°, −70.4354°] longitude and [41.0864°, 41.3386°] latitude, and in time steps of 30 min. The velocity field is reconstructed from HFR measurements as described in ref. [11], and is available on a uniform grid within the hatched polygon in Fig. 3a (Supplementary Methods and Supplementary Fig. 3).

To mimic objects fallen in the water, we use 68 Coastal Ocean Dynamics Experiment (CODE) drifters (Supplementary Methods and Fig. 3) whose GPS-tracked locations (white dots) are recorded once every 5 min. Drifters of the same design are routinely used by the U.S. Coast Guard in SAR operations. The starting time of our analysis is the 4th of August 2014 at 17:00 EDT when drifters are located close to the Muskeget channel (Fig. 3a). Figure 4a shows a zoomed version of the black square inset in Fig. 3a, along with drifter positions and the instantaneous

streamlines of the HFR velocity. We then compute TRAPs every 30 min with the updated velocity field. As expected, we find that the emergence of strong TRAPs at 19:00 (Fig. 4b) promptly organize the drifters into one-dimensional structures along TRAPs within 2 h (Fig. 4a–c). The time evolution of drifter positions along with TRAPs and velocity streamlines is available as Supplementary Movie 1. As an aggregate measure of attraction to TRAPs over time, we consider the averaged distance of each drifter from the closest TRAP (Table 1). Within one hour, from 18:00 to 19:00, the average drifter-to-TRAP distance decreases by about 30%; within 2 h, drops by about 60%. The standard deviation of the distance to TRAPs also decreases progressively, reflecting the change of an initially spread-out drifter distribution into an organized one along the TRAPs.

**Table 1 2014 field experiment.**

| Time | 18:00 | 19:00 | 20:00 |
|---|---|---|---|
| $\langle d \rangle$ [km] | 1.7 | 0.86 | 0.55 |
| sd [km] | 1.5 | 0.4 | 0.2 |

Average distance ($\langle d \rangle$) and standard deviation (sd) of drifters from the closest TRAP for the 2014 experiment (Fig. 4).

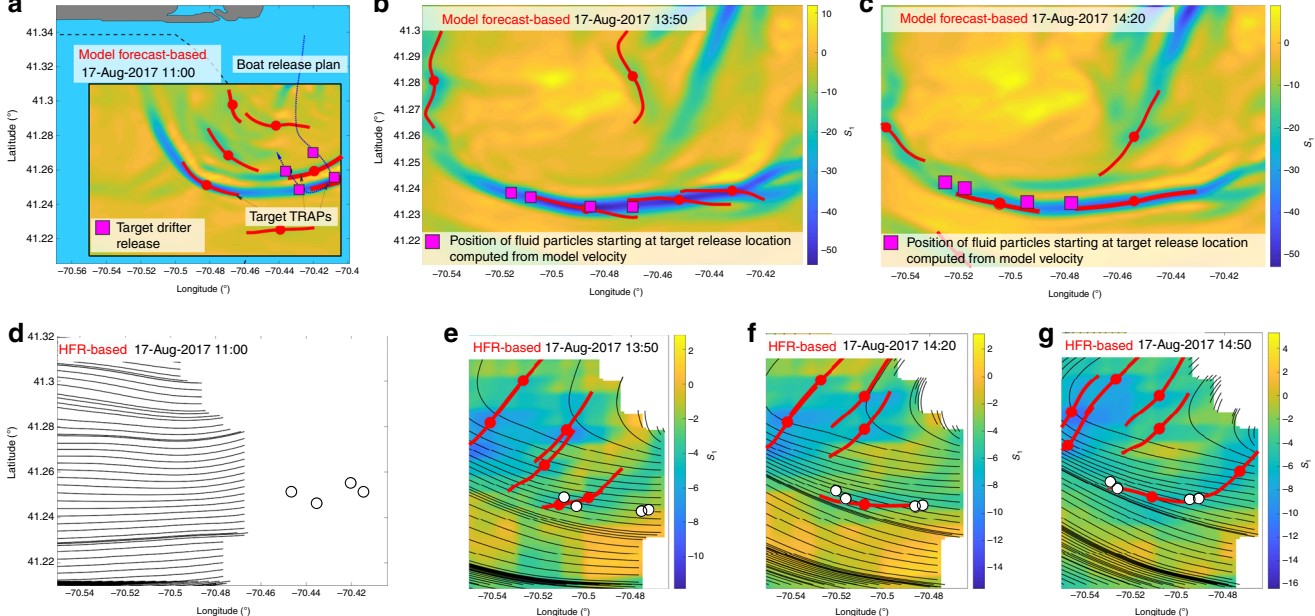

**Fig. 5 2017 experiment. a** Drifter release plan on the 17th of August 2017 based on the 24h forecast model flow velocity provided on the 16th at 7 pm. Magenta squares denote the target drifter release location at 11:00 am (EDT). **b** and **c** TRAPs from model velocity at later times in the focus region bounded by the black rectangle in **a**. Magenta squares are the current position of fluid particles starting at the drifters release location, and computed by integrating the model fluid velocity. **d** Deployed drifters based on **a**–**c** along with HFR-based TRAPs. GPS-tracked drifters position (white dots) at 11:00 am along with the flow streamlines computed from the HFR velocity field. The domain corresponds to the focus region in the panels above. **e**–**g** Drifter positions a few hours later, along with the corresponding TRAPs and streamlines computed from the HFR velocity field. The unit of $s_1$ is day$^{-1}$.

Over longer time scales (approximately a week), drifter accumulation on the ocean surface has been identified with regions of negative horizontal divergence[12,13]. The horizontal divergence diagnostic, however, can lead to both false positives and negatives: examples of particle accumulation in regions of zero or positive divergence are given in the Supplementary Methods and Supplementary Fig. 2. This is precisely the case with TRAP A in Fig. 4d, which attracts drifters strongly, even though it is located in a region of positive horizontal divergence. The negative $s_1$ values along TRAP A (Fig. 4c), by contrast, correctly predict its attraction property. Furthermore, regions of negative horizontal divergence, irrespective of their validity, tend to be large open sets (Fig. 4d), as opposed to specific, one-dimensional curves over which we observe drifters clustering. These results show that TRAPs may be completely hidden in instantaneous streamline and horizontal divergence plots, yet predict the short-term fate of passive tracers, as well as inertial objects influenced by windage, such as drifters. Although incorporating inertial, windage, and leeway effects could, in principle, provide a better prediction, in a SAR operation the inertia of the target objects is generally unknown[14] and wind information is unavailable.

Although using HFR velocity would significantly enhance the success of SAR operations[15], traditional SAR planning is generally based on model velocity data. To account for this, we conducted two more experiments to identify TRAPs from the ocean surface velocity derived from the MIT multidisciplinary simulation, estimation, and assimilation systems (MIT-MSEAS)[16], summarized in the Supplementary Methods, which assimilates local measurements, similarly to the models used in actual SAR.

For the experiment performed on the 17 August 2017, we compute TRAPs from the 24h forecast model velocity provided on the 16 August at 7 pm. We focus on a region south-east of Martha's Vineyard and identify TRAPs from 11 am on 17 August 2017 (Fig. 5a). The strongest TRAPs are located along a trench of the $s_1(\mathbf{x}, t)$ field demarcating a one-dimensional structure

containing several TRAPs with strong attraction rates. We note the presence of two parallel trenches from the model. Assuming that the real trench is somewhere in between these two because of modeling uncertainties, we released four drifters north of the lower trench (magenta squares). Figure 5b and c show later positions of fluid particles obtained by integrating the model velocity field from the target drifter release location, along with the corresponding TRAPs. The figure confirms their attracting property with respect to model data before the float deployment. Based on the release locations in Fig. 5a, Fig. 5d shows the deployed GPS-tracked drifters position (white dots) at 11:10 am within the area of interest bounded by the black rectangle in Fig. 5a, along with the streamlines computed from the HFR velocity at the same time. Panels Fig. 5e–g show later drifter positions along with the TRAPs and the streamlines computed from the HFR velocity field. Although our deployment strategy was purely based on model velocities, the comparison with actual drifter trajectories and TRAPs computed from HFR velocity shows that the model provided reliable estimates of the actual TRAPs. While these TRAPs remained hidden in streamline plots, they nevertheless attracted drifters within 3 h. Table 2 shows the average drifter-to-TRAP distance corresponding to Fig. 5e–g. Statistics at earlier times are inaccessible because drifters are in a region where the HFR-based velocity is unavailable.

**Table 2 2017 field experiment.**

| Time | 13:50 | 14:20 | 14:50 |
|---|---|---|---|
| ⟨d⟩ [km] | 1.25 | 0.4 | 0.13 |
| sd [km] | 1.1 | 0.2 | 0.1 |

Average drifter-to-TRAP distance (⟨d⟩) and standard deviation (sd) for the 2017 experiment (Fig. 5d–g).

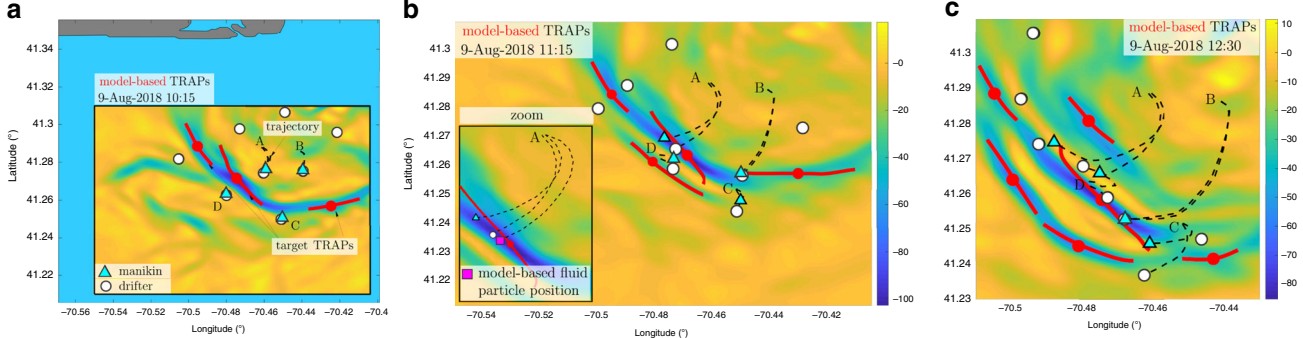

**Fig. 6 2018 experiment. a** Deployed drifters and manikins on the 9th of August 2018 based on TRAPs computed from the 24h forecast model velocity provided on the 8th at 8 pm. White dots and cyan triangles show the GPS-tracked location of CODE drifters and manikins (Fig. 3) at 10:15 am of the 9th August 2018. Dashed lines show object trajectories released at locations A–D from their deployment to the current time. **b** and **c** Drifter and manikin positions at later times, along with the corresponding model-based TRAPs. The inset in **b** shows a zoomed version of the manikin and drifter trajectories released in A, along with the trajectory of a fluid particle (magenta square) obtained by integrating the model velocity from the same initial condition of the drifter and manikin. The unit of $s_1$ is day$^{-1}$.

In our last experiment, to mimic an even more realistic SAR scenario, we considered a larger set of initially spread-out floating objects consisting of 8 CODE drifters and 4 OSCAR Water Rescue Training manikins manufactured by Emerald Marine Products (Supplementary Methods and Fig. 3b and c). Using a strategy similar to the 2017 experiment, we designed a deployment for the 9th of August 2018, based on the center forecast model velocity field provided on the 8th of August at 8 pm. Figure 6a shows the target model-based TRAPs at 10:15 am on the 9th of August 2018, along with all released drifter (white dots) and manikin (cyan triangles) positions. We show only the strongest targeted TRAPs ranked by $s_1$. Dashed curves represent the GPS-tracked trajectories of the deployed objects from their release until 10:15 am. In this experiment, we used two WHOI vessels for deployment: one for the release of drifters and manikins at the locations demarcated by A–D in Fig. 6a, and a second vessel for the remaining drifters. Figure 6b and c show the later positions of drifters and manikins along with their trajectories and the recomputed model-based TRAPs. Because of a relocation of HFR towers in 2018, HFR velocity was not available in the domain shown in Fig. 6. Similar to the previous experiments, both drifters and manikins show a striking alignment with the strongest nearby TRAPs computed from the fluid model velocity within 2 h. Table 3 shows the average drifter/manikin-to-TRAP distance corresponding to Fig. 6.

A closer inspection of the deployed drifter and manikin trajectories shows that these two different objects may follow different paths even after short times (<2 h). This is clearly the case for objects released from locations A, D, C shown in Fig. 6. In the inset of Fig. 6b, we show a zoomed version of the drifter and manikin trajectories deployed in A, together with the trajectory of a fluid particle (magenta square) obtained by integrating the model velocity from A. Even though fluid particles, drifters and manikins all follow different trajectories due to inertia, windage and other effects, they invariably converge to the same TRAP, which provides a highly robust attracting skeleton of the underlying flow. In the Supplementary Note 1, we compare TRAP predictions with trajectory-based ones typically used in SAR. We use nine ensemble velocity field forecasts arising from parametric uncertainty sources, and compute the corresponding trajectories using the experimental deployment locations as initial conditions. We find that even though drifter, manikins, and ensemble trajectories all differ from each other, they all converge to nearby TRAPs computed from the center-forecast velocity. Using simple mathematical arguments, we also show that TRAPs are intrinsically robust under uncertainties

over short times, as opposed to trajectory-based methods, whose sensitivity to uncertainties grow with the largest Lyapunov exponent of the underlying velocity field. Admittedly, TRAPs lose their predictive power over longer time scales because of their instantaneous nature. Shorter time scales (<6 h in this context), however, are precisely the relevant ones for SAR and hazard response scenarios.

## Discussion

We have predicted and experimentally verified the existence of TRAPs, which govern short-term trajectory behavior in chaotic ocean currents characterized by high uncertainties. We expect TRAPs to provide critical information in emergency response situations, such as SAR and oil spill containment, in which operational decisions need to be made quickly about optimal resource allocation. Existing SAR techniques handle uncertain parameters in models of floating objects by averaging several Monte Carlo Simulations and providing probability maps for the objects' location. These maps, however, are not readily interpretable for practical use and can converge slowly due to the underlying chaotic processes. TRAPs and their attraction rates, by contrast, are easily interpretable and highly localized curves which can be computed and updated instantaneously from snapshots of the ocean surface velocity. This eliminates the need for costly trajectory calculations and yields fast input for search-asset allocation.

We have emulated different SAR scenarios in three ocean field experiments carried out south of Martha's Vineyard. We computed TRAPS both from HFR submesoscale ocean surface velocity and from model velocities similar to those available in SAR operations. Our results indicate that TRAPs have significant predictive power in assessing the most likely current positions of objects and people fallen in water at uncertain locations. We have specifically found that TRAPs invariably attract nearby floating objects within two-to-three hours, even though they remain hidden to instantaneous streamlines and horizontal divergence

**Table 3 2018 field experiment.**

| Time | 10:15 | 11:15 | 12:30 |
|---|---|---|---|
| ⟨d⟩ [km] | 1.9 | 0.8 | 0.4 |
| sd [km] | 1 | 0.8 | 0.5 |

Average drifter/manikin-to-TRAP distance (⟨d⟩) and standard deviation (sd) for the 2018 experiment shown in Fig. 6.

fields, which also rely on the same Eulerian velocity input. Such a short timing is critical in SAR, as after 6 h, the likelihood of rescuing people alive drops significantly. We therefore envision that sea TRAPs will enhance existing SAR techniques, providing critical information to save lives and limit the fall-out from environmental disasters during hazard responses.

## Methods

### Algorithm 1.
Compute TRAPs in two-dimensional flows[8]

---

**Input:** A two-dimensional velocity field $\mathbf{v}(\mathbf{x}, t)$

1. Compute the Jacobian of the velocity field $\nabla\mathbf{v}$ by numerically differentiating $\mathbf{v}$ with respect to $\mathbf{x}$, and the rate-of-strain tensor $\mathbf{S}(\mathbf{x}, t) = \frac{1}{2}(\nabla\mathbf{v}(\mathbf{x}, t) + [\nabla\mathbf{v}(\mathbf{x}, t)]^*)$ at the current time $t$ on a grid over the $\mathbf{x} = (x_1, x_2)$ coordinates, where $*$ denotes matrix transposition.

2. Compute the smallest eigenvalue field $s_1(\mathbf{x}, t) \leq s_2(\mathbf{x}, t)$ and the unit eigenvector field $\mathbf{e}_2(\mathbf{x}, t)$ of $\mathbf{S}(\mathbf{x}, t)$ associated to $s_2(\mathbf{x}, t)$.

3. Compute the set $\mathcal{S}_m(t)$ of negative local minima of $s_1(\mathbf{x}, t)$.

4. Compute TRAPs as solutions of the ODE
$$\begin{cases} \mathbf{r}'(\tau) = \text{sign}\langle\mathbf{e}_2(\mathbf{r}(\tau)), \mathbf{r}'(\tau - \Delta)\rangle\mathbf{e}_2(\mathbf{r}(\tau)) \\ \mathbf{r}(0) \in \mathcal{S}_m, \end{cases}$$
where $\tau$ denotes the arclength parameter, $'$ differentiation with respect to $\tau$, and $\Delta$ the arclength increment between two nearby points on the TRAP. Stop integration when $s_1(\mathbf{r}(\tau)) > 0.3s_1(\mathbf{r}(0))$ or $s_1(\mathbf{r}(s)) \geq 0$.

**Output:** TRAPs at time $t$ along with their normal attraction rate field $s_1(\mathbf{x}, t)$.

---

The sign term in step 4 guarantees the local smoothness of the direction field $\mathbf{e}_2$, and the termination conditions ensure that the attraction rate of subsets of TRAPs is at least 30% of the core attraction rate, hence exerting a distinguished attraction compared to nearby structures.

In the Supplementary Methods, we show the robustness of TRAPs under uncertainties, as well as why TRAPs remain hidden to the widely used horizontal divergence field. In the Supplementary Methods, we also describe in detail our HFR velocity and model velocity datasets, as well as drifter and manikin datasets.

## Data availability
All data are available to individual researchers upon request from the corresponding authors.

## Code availability
All codes are available to individual researchers upon request from the corresponding authors.

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

## Acknowledgements
We are grateful to Margaux Filippi, Michael Allshouse, Javier González-Rocha, Peter Nolan, Siavash Ameli, Patrick Haley Jr., and the MSEAS team for their contribution in the field experiments. We also acknowledge the NSF Hazard funding grant no. 1520825. S.D.R. acknowledges support from NSF grant no. 1821145. M.S. would like to acknowledge support from the Schmidt Science Fellowship (https://schmidtsciencefellows.org/). G.H. acknowledges support from the Turbulent Super-structures Program of the German National Science Foundation (DFG).

## Author contributions
M.S. and G.H. designed research; M.S. and P.S. performed research; I.R. provided the drifter data; A.K. provided the HFR velocity data, S.R. provided the manikin data, P.L. provided the model velocity data, A.A. provided knowledge and expertise in SAR; T.P. and I.R. led the field experiments. All authors contributed to the field experiments. M.S., G.H., T.P., and P.S. wrote the paper.

## Competing interests
The authors declare no competing interests.
