## [Peer Review File · Nature Communications]

Reviewers' comments:

Reviewer #1 (Remarks to the Author):

This paper addresses the question of how to analyze ocean velocity fields so as to predict where floating objects will accumulate.

The paper tests a recent interesting theoretical result that for steady 2D velocity field, particles moving with the flow will accumulate along lines of the locally attracting strain eigenfunction (TRAPS). The paper claims to demonstrate this effect using measurements in a coastal environment.

The demonstration rests on three field experiments in which the velocity field was measured and floating objects were tracked for a few hours. The claim is that these objects "were attracted" to the TRAP lines.

More specifically, the experiments aim to test whether the green dots (drifting objects) in Figures 4bc and 5b move toward the red lines (TRAPS computed from measured velocity fields) in the same figures. No statistical tests are given, just these pictures.

I can formulate a simple statistical test for 'attraction', namely that the dots lie on top of the TRAPS lines by the end of the measurements. There are roughly 70 green dots. By the end, about 6 dots are on the lines. Thus 91% of the drifting object do not end up on the lines, falsifying the hypothesis. The paper claims that these data support the hypothesis.

I suspect that there really is something to this idea and that a more careful analysis would yield a more positive result. What is needed is a quantitative measure of the 'attraction' along with a statistical analysis of whether the data shows attraction relative to a null hypothesis.

Clearly a "major revisions required" in the review language of other journals.

Reviewer #2 (Remarks to the Author):

In this manuscript, the authors introduce a more efficient approach to focus Search and Rescue (SAR) efforts that is based on Objective Eulerian Coherent Structures. With this technique the authors claim to retain the advantages of eulerian methods and improve on many aspects of traditional SAR.

The manuscript is very well written and I believe the results will be of interest to others in the community and the wider field. I only have a few minor comments and suggestions mostly in the spirit of clarification. Thus, I recommend publication after minor revision. Here's a list of specific comments:

- In some parts of the text the authors comment that their approach is faster than traditional approaches but don't mention how much faster. There are comparisons of robustness and domain limitations in the SIs, so I think it wouldn't be hard to include a sentence making a comparison of computing time between methods.

- line 81: when discussing the sensitivity to release time, location, etc., please follow it up with a reference for further reading.

- line 91: "short-term limit" of LCSs is a term that isn't very intuitive to the reader at this point (even though it becomes clearer later). I suggest changing/explaining this.
- lines 108-112: The sentences are a bit convoluted and wordings like "theoretical centerpieces", for example, deteriorate clarity. This is especially an issue if you're introducing new terms to the reader, as is the case here. Furthermore, the text points to figure 2a which has e1 and e2 in it, neither of which has been explained to this point, adding to the confusion. So I suggest re-writing this part in a simpler, clearer manner.
- line 124: This is a very interesting feature and it is not straightforward to intuitively understand. Thus I think the authors should give the reader resources to visualize this. Exploring this further in a SI (similar to SI.3, which explored divergence) would be a good idea, but I also think an animation would be very helpful (plotting the evolution of drifters along with TRAPs and the horizontal divergence as a colormap). If the authors think that neither of these options is within their scope, I think they could at least reference the animation in Serra & Haller (Chaos, 2016), which serves this purpose reasonably well.
- line 128: replace "SAR" with "traditional SAR techniques", to make it clearer.
- Calls to supplementary information throughout the text do not specify which section of the SI material. Please include that information throughout the paper for the ease of the reader.
- Replace "divergence" with "horizontal divergence" throughout the text for the sake of clarity.
- line 174: replace "SAR" with "traditional SAR".
- Please make the color ranges in panels A of Figure 5 the same (something like -50 to 10). Please do the same for panels B (a different range, of course). Also please make the x-y axis ranges for all panels in Fig 5B the same. At the moment the left panel has a different range and no colormap, which makes it look like it's showing something different.
- Please also make the color ranges in Fig 6 uniform.
- line 229: What is meant here by "shorter time scales"?
- line 330: add the acronym "QC" after quality controlled since that acronym is used later.
- line 340: By numerical recipes do you mean William Press' book? If so, please provide citation with the proper pages of the algorithm.
- It may be interesting to include a sentence or two about the horizontal scale of SAR operations. How big of a domain generally needs to be considered? How does this size compare with the usual size of TRAPs?
- Is the schematic in the inset of Fig 1 purely for illustrative purposes? Or are there reasons to place a TRAP where it was placed?

- I'm not familiar with SAR operations, but it may be relevant to include a sentence about how the active swimming of people may complicate the picture (or how it is negligible).

- It seems to me that (i) the attraction in divergent regions and (ii) the fact that TRAPs are hidden from streamlines are related. If that is true, I think it's worth including a couple of sentences about this.

Reviewer #3 (Remarks to the Author):

Review of NCOMMS-19-30144-T by Serra et al.

The authors investigate a novel method for nautical search-and-rescue based on the least eigenvalue of the strain tensor of the surface horizontal current field. They show that the method compares well with an alternative method based on the convergence of the surface current, and that it can be used in conjunction with either regional scale ocean models or high-frequency radar measurements of the surface current. This is a first-rate piece of work and is well suited to publication in Nature.

As a good paper should, this paper raises many interesting questions. I will list a few in hopes that the authors may be able to address them within the scope of the current paper. This is in no way a precondition for approving publication. I am struck by the happy coincidence between the attraction time scale $1/s_1$ and the typical human survival time at sea, i.e. a few hours. What governs $1/s_1$? It must be longer in weakly strained regions such as mesoscale eddy cores. Is it generally sensitive to distance from shore, or proximity to major currents like the Gulf Stream? Is it sensitive to the resolution used to approximate the derivatives of the velocity field? Along with variations in survival time (e.g. due to water temperature), perhaps these factors could define circumstances in which use of TRAPs would be recommended versus other search methods.

The phrase "under uncertainty" (lines 65, 140, 225, 291, 302) seems cryptic to me. Is there a more explicit way to say it?

Line 136: Change "its" to "their".

Figure 4d: The arrow indicating TRAP A is very hard to see.

Line 227: Change "loose" to "lose".

I recommend publication after consideration of the comments above.

Response to Reviewer 1 of Search and rescue at sea aided by hidden flow structures

January 14, 2020

Reviewer's comments are in boldface, our responses are in regular font and changes in the manuscript are in red.

This paper addresses the question of how to analyze ocean velocity fields so as to predict where floating objects will accumulate. The paper tests a recent interesting theoretical result that for steady 2D velocity field, particles moving with the flow will accumulate along lines of the locally attracting strain eigenfunction (TRAPS). The paper claims to demonstrate this effect using measurements in a coastal environment. The demonstration rests on three field experiments in which the velocity field was measured and floating objects were tracked for a few hours. The claim is that these objects 'were attracted' to the TRAP lines. More specifically, the experiments aim to test whether the green dots (drifting objects) in Figures 4bc and 5b move toward the red lines (TRAPS computed from measured velocity fields) in the same figures. No statistical tests are given, just these pictures. I can formulate a simple statistical test for 'attraction', namely that the dots lie on top of the TRAPs lines by the end of the measurements. There are roughly 70 green dots. By the end, about 6 dots are on the lines. Thus 91% of the drifting object do not end up on the lines, falsifying the hypothesis. The paper claims that these data support the hypothesis. I suspect that there really is something to this idea and that a more careful analysis would yield a more positive result. What is needed is a quantitative measure of the 'attraction' along with a statistical analysis of whether the data shows attraction relative to a null hypothesis.

We appreciate the Reviewer's comment. In the revised manuscript, we have quantified the averaged distance of each drifter to the closest TRAP over time for the three field experiments. We have invariably observed that the mean drifter/manikin-to-TRAP distance decreases considerably within a few hours. Similarly, we have also found a considerable decrease in the standard deviation of the distance confirming the organizing role of TRAPs on floating objects. The decay in the object-to-TRAP distance and the reduction in the object-spread make TRAPs promising candidates for optimal search paths in SAR at sea, as illustrated in Fig. 1.

1. To convey the above in the text, we have modified the first paragraph of section 2 as follows.

As an aggregate measure of attraction to TRAPs over time, we consider the averaged distance of each drifter from the closest TRAP (Table 1). Within one hour, from 18:00 to 19:00, the average drifter-to-TRAP distance decreases by about 30%; within two hours, drops by about 60%. The standard deviation of the distance to TRAPs also decreases progressively, reflecting the change of an initially spread-out drifter distribution into an organized one along the TRAPs.

time	18:00	19:00	20:00
$\langle d \rangle$ [km]	1.7	0.86	0.55
sd [km]	1.5	0.4	0.2

Table 1: Average distance ($\langle d \rangle$) and standard deviation (sd) of drifters from the closest TRAP for the 2014 experiment (Fig. 4).

2. We have modified the fifth paragraph of section 2 as follows.

Table 2 shows the average drifter-to-TRAP distance corresponding to Fig. 5b. Statistics at earlier times are inaccessible because drifters are in a region where the HFR-based velocity is unavailable.

time	13:50	14:20	14:50
$\langle d \rangle$ [km]	1.25	0.4	0.13
sd [km]	1.1	0.2	0.1

Table 2: Average drifters-TRAPs distance ($\langle d \rangle$) and standard deviation (sd) for the 2017 experiment (Fig. 5b).

3. We have modified the sixth paragraph of section 2 as follows.

Table 3 shows the average drifter/manikin-to-TRAP distance corresponding to Fig. 6.

time	10:15	11:15	12:30
$\langle d \rangle$ [km]	1.9	0.8	0.4
sd [km]	1	0.8	0.5

Table 3: Average drifter/manikin-to-TRAP distance ($\langle d \rangle$) and standard deviation (sd) for the 2018 experiment shown in Fig. 6.

The purpose of this work is to introduce and experimentally validate a new tool to tackle a SAR scenario. Within the scope of our experiments, we have found that the statistics above confirm the key role of TRAPs on floating objects. We are grateful to the Referee for his/her suggestion to perform this analysis. We are planning to conduct more in-depth statistical analyses in future works.

Response to Reviewer 2 of Search and rescue at sea aided by hidden flow structures

January 14, 2020

Reviewer's comments are in boldface, our responses are in regular font and changes in the manuscript are in red.

In this manuscript, the authors introduce a more efficient approach to focus Search and Rescue (SAR) efforts that is based on Objective Eulerian Coherent Structures. With this technique the authors claim to retain the advantages of eulerian methods and improve on many aspects of traditional SAR. The manuscript is very well written and I believe the results will be of interest to others in the community and the wider field. I only have a few minor comments and suggestions mostly in the spirit of clarification. Thus, I recommend publication after minor revision. Here's a list of specific comments:

We appreciate the Reviewer's positive comments.

- In some parts of the text the authors comment that their approach is faster than traditional approaches but don't mention how much faster. There are comparisons of robustness and domain limitations in the SIs, so I think it wouldn't be hard to include a sentence making a comparison of computing time between methods.

We addressed this comment by adding a new section in the SI as reported below.

SI.4 Computational times in SAR

Here we provide a rough estimate of the computational time gain of TRAP-based SAR planning compared to existing techniques. A Search and Rescue Optimal Planning System (SAROPS) consists of approximately twelve steps¹. About 70% of the total computational time is required for downloading the data from the Environmental Data Server (EDS) to the SAROPS computers and for generating search patterns from the probability-distribution maps for the lost object's location. High-performance SAROPS computers are mainly needed to perform Monte Carlo simulations and for computing search paths from the 2D probability maps. The robustness under uncertainties and the Eulerian nature of TRAPs, which are one-dimensional curves suitable for search-asset allocation, eliminate the need for Monte Carlo simulations and further post-processing of search patterns. TRAPs, therefore, could be potentially computed at the EDS-level, reducing the computational time by at least 50%, or at least providing valuable additional information to existing strategies at no extra costs.

- line 81: when discussing the sensitivity to release time, location, etc., please follow it up with a reference for further reading.

¹http://rdept.cgaux.org/documents/ManualsTemp/USCG_SAR_Addendum.pdf

We have provided a reference as reported below.

One such set of factors is the exact release time, release location and length of the observation period [9]².

- line 91: "short-term limit" of LCSs is a term that isn't very intuitive to the reader at this point (even though it becomes clearer later). I suggest changing/explaining this.

We have modified the sentence as follows.

An alternative to these Lagrangian approaches is to find the **instantaneous limits** of LCSs purely from Eulerian observations, thereby avoiding all the pitfalls of trajectory integration. These limiting (i.e. **infinitesimally-small-advection-time**) LCSs, predict pathways and barriers to short-term material transport until the next batch of updated velocity information becomes available.

- lines 108-112: The sentences are a bit convoluted and wordings like "theoretical centerpieces", for example, deteriorate clarity. This is especially an issue if you're introducing new terms to the reader, as is the case here. Furthermore, the text points to figure 2a which has e1 and e2 in it, neither of which has been explained to this point, adding to the confusion. So I suggest re-writing this part in a simpler, clearer manner.

We have modified the sentence as follows.

As such, OECSs are extensions of the notions of unstable (and stable) manifolds of a saddle point in a steady flow, which attract (and repel, respectively) fluid elements and hence ultimately serve as the **backbones** of deforming tracer patterns. **In unsteady flows and over short times, however, saddle stagnation points lose their connection with material transport [8]³. Instead, objective saddle points – the cores of hyperbolic OECSs – emerge, with associated attracting and repelling OECSs (Fig. 2a).** In our present context, we will refer to attracting OECSs and objective saddle points as *TRAPs* and *TRAP cores*.

We have also modified the caption of Fig. 2 as follows.

(a) Deformation of a fluid patch close to an objective saddle point in an unsteady flow. Over short times, a fluid patch aligns with the repelling OECS, and squeezes along the attracting OECS, which both evolve over time. Attracting (Repelling) OECSs are everywhere tangent to **the eigenvector field e_2 (e_1) of the rate-of-strain tensor, with their cores located at minima of the smallest eigenvalue λ_1 (λ_2).**

- line 124: This is a very interesting feature and it is not straightforward to intuitively understand. Thus I think the authors should give the reader resources to visualize this. Exploring this further in a SI (similar to SI.3, which explored divergence) would be a good idea, but I also think an animation would be very helpful (plotting the evolution of drifters along with TRAPs and the horizontal divergence as a colormap). If the authors think that neither of these options is within their scope, I think they could at least reference the animation in Serra & Haller (Chaos, 2016), which serves this purpose reasonably well.

We have addressed this comment by adding a supplementary movie SMovie 1, and modifying the paragraph mentioned by the Referee as follows.

Remarkably, such pathways remain generally hidden in streamline plots, can even be perpendicular to streamlines (Figs. 2b,4 and SMovie1), and exist also in divergence-free flows, as shown in Fig. 11 and the corresponding movie in [8]. See [8] for a detailed explanation.

- line 128: replace "SAR" with "traditional SAR techniques", to make it clearer.

We have addressed this comment.

- Calls to supplementary information throughout the text do not specify which section of the SI material. Please include that information throughout the paper for the ease of the reader.

²Reference [9] corresponds to reference [1] in this response letter.

³Reference [8] corresponds to reference [2] in this response letter.

We have addressed this comment.

- Replace "divergence" with "horizontal divergence" throughout the text for the sake of clarity.

We have addressed this comment.

- line 174: replace "SAR" with "traditional SAR".

We have addressed this comment.

- Please make the color ranges in panels A of Figure 5 the same (something like -50 to 10). Please do the same for panels B (a different range, of course). Also please make the x-y axis ranges for all panels in Fig 5B the same. At the moment the left panel has a different range and no colormap, which makes it look like it's showing something different. Please also make the color ranges in Fig 6 uniform.

We believe that an automatic color bar range shows better the evolution of attraction rates over time. Setting a fixed color range over time, by contrast, would hide such attracting changes in different color intensities that are harder to read. For consistency between the model forecast-based release and the HFR-based computation, the axis range of Fig. 5b (left) corresponds to those in Fig. 5a (left), as described in the caption. We prefer to keep these the same also to connect the GPS-based drifter positions (Fig. 5b left) with the boat release plan (Fig. 5a left) as well as to illustrate that right after the release drifters are in a domain where HFR-velocities are unavailable.

- line 229: What is meant here by "shorter time scales"?

We have modified the sentence as follows:

Shorter time scales (**less than 6 hours in this context**), however, are precisely the relevant ones for SAR and hazard response scenarios.

- line 330: add the acronym "QC" after quality controlled since that acronym is used later.

We have addressed this comment.

- line 340: By numerical recipes do you mean William Press' book? If so, please provide citation with the proper pages of the algorithm.

We have addressed this comment by adding the appropriate references as follows.

This estimate uses the radial velocity error estimates (the weighted standard deviation of the individual HF radar radial returns found within each 5 degree azimuthal bin average) in a standard vector error calculation [24,25,26]⁴.

- It may be interesting to include a sentence or two about the horizontal scale of SAR operations. How big of a domain generally needs to be considered? How does this size compare with the usual size of TRAPs?

We are unaware of general recipes for a priori estimating the domain and TRAP sizes in SAR scenarios. We have addressed this comment by adding a sentence at the end of SI.5.2 as follows.

In a SAR operation, the extent of the search domain evolves over time, starting from the last seen location area and its associated uncertainties. The domain then grows according to the overall Lagrangian transport and its total extent is only constrained by the shoreline of the body of water where the case occurs. TRAPs, therefore, can be computed within this evolving domain, readily providing regularly updated optimal search paths. Finally, we note that the test area off Martha Vineyard analyzed here would be typical for a first-day search.

- Is the schematic in the inset of Fig 1 purely for illustrative purposes? Or are there

⁴References [24,25,26] correspond to [3, 4, 5] in this response letter

reasons to place a TRAP where it was placed?

The Reviewer is correct. We have clarified this point by made this point by modifying the caption of Fig. 2 as follows:

(c) **Illustration** of a TRAP evolving in time and attracting within a few hours floating objects whose uncertain initial locations are represented by a square set of green dots.

- I'm not familiar with SAR operations, but it may be relevant to include a sentence about how the active swimming of people may complicate the picture (or how it is negligible).

Active swimmers are relatively rare in SAR scenarios because a large percentage of the population lack efficient swimming techniques and training. As of now, SAROPS does not account for active swimming. Although it is possible to modify TRAP predictions in the case of active swimmers, we prefer to keep our manuscript for general SAR conditions.

- It seems to me that (i) the attraction in divergent regions and (ii) the fact that TRAPs are hidden from streamlines are related. If that is true, I think it's worth including a couple of sentences about this.

The Reviewer may well be right, but we are unaware of any rigorous and general argument that would relate (i) to (ii), so we would rather not speculate.

References

- [1] G. Haller. Lagrangian coherent structures. *Annual Rev. Fluid. Mech.*, 47:137–162, 2015.
- [2] M. Serra and G. Haller. Objective Eulerian coherent structures. *Chaos*, 26(5):053110, 2016.
- [3] DE Wells, N Beck, D Delikaraoglou, A Kleusberg, E Krakiwsky, G Lachapelle, R Langley, M Nakiboglu, KP Schwarz, J Tranquilla, et al. Guide to gps positioning, 2nd printing with corrections. *Canadian GPS Associates, Fredericton NB*, 1987.
- [4] Rick D Chapman and Hans C Graber. Validation of hf radar measurements. *Oceanography*, 10(2):76–79, 1997.
- [5] Belinda Lipa. Uncertainties in seasonde current velocities. In *Proceedings of the IEEE/OES Seventh Working Conference on Current Measurement Technology, 2003.*, pages 95–100. IEEE, 2003.

Response to Reviewer 3 of Search and rescue at sea aided by hidden flow structures

January 14, 2020

Reviewer's comments are in boldface, our responses are in regular font and changes in the manuscript are in red.

The authors investigate a novel method for nautical search-and-rescue based on the least eigenvalue of the strain tensor of the surface horizontal current field. They show that the method compares well with an alternative method based on the convergence of the surface current, and that it can be used in conjunction with either regional scale ocean models or high-frequency radar measurements of the surface current. This is a first-rate piece of work and is well suited to publication in Nature. I recommend publication after consideration of the comments below.

We appreciate the Reviewer's positive comment.

As a good paper should, this paper raises many interesting questions. I will list a few in hopes that the authors may be able to address them within the scope of the current paper. This is in no way a precondition for approving publication. I am struck by the happy coincidence between the attraction time scale $1/s1$ and the typical human survival time at sea, i.e. a few hours. What governs $1/s1$? It must be longer in weakly strained regions such as mesoscale eddy cores. Is it generally sensitive to distance from shore, or proximity to major currents like the Gulf Stream? Is it sensitive to the resolution used to approximate the derivatives of the velocity field?

The Reviewer is correct, $1/s1$ is larger in more quiescent regions of the ocean and smaller close to shore or energetic flow features. When using sensed velocities, the derivatives of the velocity field should be approximated numerically taking into consideration the resolution of the velocity field to ensure consistency. We discuss this point at the end of the SI.5.1, as reported below.

Finally, we note that HFR velocities are computed by averaging the raw radial velocity estimates with a 800m window radius at each grid point, spaced 800m apart from each other [23]¹. To yield an accurate computation of TRAPs from HFR velocities consistent with the way the data are processed, we smooth (\mathbf{x}, t) with a spatial average filter whose width corresponds to two grid sizes (1600m), as in [27]².

Along with variations in survival time (e.g. due to water temperature), perhaps these factors could define circumstances in which use of TRAPs would be recommended versus other search methods.

We have addressed this comment by adding a new section in the SI as reported below.

¹Reference [23] corresponds to reference [1] in this response letter.

²Reference [27] corresponds to reference [2] in this response letter.

SI.4 Computational times in SAR

Here we provide a rough estimate of the computational time gain of TRAP-based SAR planning compared to existing techniques. A Search and Rescue Optimal Planning System (SAROPS) consists of approximately twelve steps³. About 70% of the total computational time is required for downloading the data from the Environmental Data Server (EDS) to the SAROPS computers and for generating search patterns from the probability-distribution maps for the lost object's location. High-performance SAROPS computers are mainly needed to perform Monte Carlo simulations and for computing search paths from the 2D probability maps. The robustness under uncertainties and the Eulerian nature of TRAPs, which are one-dimensional curves suitable for search-asset allocation, eliminate the need for Monte Carlo simulations and further post-processing of search patterns. TRAPs, therefore, could be potentially computed at the EDS-level, reducing the computational time by at least 50%, or at least providing valuable additional information to existing strategies at no extra costs.

The phrase 'under uncertainty' (lines 65, 140, 225, 291, 302) seems cryptic to me. Is there a more explicit way to say it?

To our knowledge, 'robustness under uncertainties' and 'sensitivities to uncertainties' are the technical terms used in the language of stochastic processes and also adopted in the SAR literature.

Line 136: Change 'its' to 'their'.

We have addressed this comment.

Figure 4d: The arrow indicating TRAP A is very hard to see.

We have modified Fig. 4d doubling the arrow width.

Line 227: Change 'loose' to 'lose'.

We have addressed this comment.

References

- [1] A. Kirincich. Remote sensing of the surface wind field over the coastal ocean via direct calibration of HF radar backscatter power. *J. Atmosph. and Oceanic Tech.*, 33(7):1377–1392, 2016.
- [2] A. Kirincich. The occurrence, drivers, and implications of submesoscale eddies on the Martha's Vineyard inner shelf. *J. Phys. Oceanogr.*, 46(9):2645–2662, 2016.

³http://rdept.cgaux.org/documents/ManualsTemp/USCG_SAR_Addendum.pdf

REVIEWERS' COMMENTS:

Reviewer #1 (Remarks to the Author):

I asked for quantitative estimates of the convergence of drifting objects to the TRAPS. The new manuscript contains these and is thus suitable for publication in its present form.